# Research on the evolution and evaluation of Chinese sports industry policies based on intelligent analysis

Zhaoyang Pan[1], Liang Ma[2]*

1 Sports Department, China Women's University, Beijing, China, 2 Library, China Women's University, Beijing, China

* maliang@cwu.edu.cn

## Abstract

In recent decades, with the support and traction of a number of key policy steps, the scale of China's sports industry has achieved a new leap. The optimization of industrial structure has made new progress. From "nascent" to "strong", China's sports industry grows in importance of the national economy. In the meantime, sport is a significant way to promote health. With the rapid growth of people's requirements for sport and health, it is urgent to re-evaluate the past development path and formulate new directions so as to continuously improve and optimize the system. This study systematically sorts out China's sports industry documents at different stages, and describes the focus of each stage and the overall evolution track. On this basis, text mining and quantitative evaluation being used to extract high-frequency words of sports industry documents, and a sports industry document evaluation system including 9 first-level indicators and 47 second-level indicators is established. In this study, text similarity analysis is used to realize intelligent PMC index analysis, which effectively improves the analysis efficiency and makes up for the deficiency of simple qualitative analysis. According to the study, China's sports industry policies are scientific and effective. Combining with the development direction of industrial transformation, it provides ideas for the future adjustment and optimization of sports industry evolution path.

## 1. Introduction

As a green and sunrise industry, the sports industry takes a crucial part in tapping and releasing the consumption potential of residents, cultivating new economic development points and enhancing new pull force for economic progress. With the arrival of the new era, sport is no longer limited to the traditional competitive arena, but deeply integrated with the economy, culture, environment, technology and other fields, showing unprecedented vitality and potential. The development of China's sports industry not only reflects the law of its own evolution, but also has distinct Chinese characteristics and characteristics of the times. As one of the vital pillars for national economic development, the status of sports industry has become increasingly prominent. It involves a variety of industrial forms, has the particularity of multi-

**Data Availability Statement:** All relevant data are within the manuscript and its Supporting Information files.

**Funding:** The author(s) received no specific funding for this work.

**Competing interests:** The authors have declared that no competing interests exist.

industry integration, and has a stronger dependence on policies. Meanwhile, sports industry plays an important role in the development of national economy, and its innovation and development are inseparable from the precise guidance of national policies. Sports industry policy is an important means of national sports industry governance, a development line and action criteria determined by the government to achieve the progress goals, and an important basis and means for the state to optimize and supervise the evolution. Quantitative policy analysis can reflect the government's emphasis on the industry, and policy text analysis can enhance the predictability and direction of decision-making, clarify the conditions for policy implementation, and ensure that the policy can achieve the expected effect in the implementation process and adapt to the changing environment. In recent years, Chinese government has successively issued a variety of sports industry policies at the national level, and gradually built a relatively complete sports industry policy system. In order to solve the difficulties and blocking points of industrial development, a series of policy innovations and mechanism innovations have been carried out, and the main goals and tasks of high-quality development of sports industry have been clarified. Various local sports industry policy tools are also equipped for macro-control and governance. In this study, Documents refer specifically to policy documents.

As can be seen from Fig 1, China's sports industry indicators are improving year by year. By 2022, the total scale of China's sports industry has jumped to 3.3 trillion yuan, and the added value has reached 1.3 trillion yuan, an increase of 5.9% and 6.9% respectively, far exceeding the GDP growth rate in the same period. Since 2012, the average annual growth rate of the total scale and added value of China's sports industry has been as high as 13.2% and 15.4%, respectively, and the proportion of added value of China's sports industry in GDP has climbed to 1.08%. The sports industry is gradually becoming a new blue ocean for China's economic growth.

This study uses the ways of intelligent bibliometric and content analysis to study the sports industry policy in China. Through the application of intelligent algorithm, realizes the automatic calculation of PMC index, high efficiency processing. The intelligent analysis method such as PMC index evaluating text mining can avoid the subjectivity of manual judgment and make policy evaluation more timely. By studying the evolution of policy formulation and adjustment, the comprehensive quantitative evaluation of China's sports industry policy text

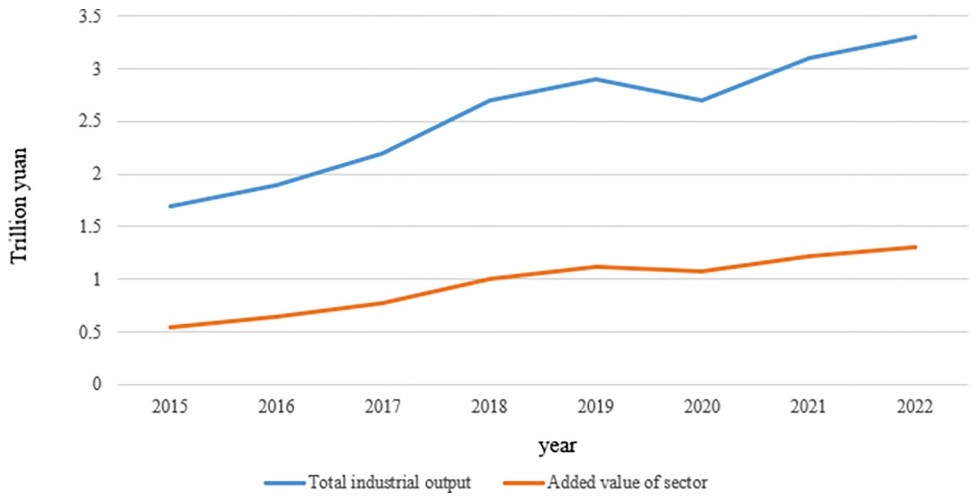

**Fig 1. Sports industry development economic indicators situation chart.**

moreover deepens the understanding of the evolution of China's sports industry, it provides reference for future sports industry policy formulation.

## 2. Literature review

### 2.1 Research on policy assessment means

Policy assessment is a factual evaluation process of scientific measurement of policy effectiveness and implementation process, as well as a value evaluation process of judging the benefit distribution and welfare changes generated by policies on social economy [1]. Process comparison method, value analysis method and questionnaire survey is often used in policy assessment, and there are two main categories. The first species is based on econometric model to analyze the impact of a certain polices on the variables related to economy and society by quantitative or qualitative methods. It includes the top-down model represented by FEEM-CMCC model, the bottom-up technical model represented by Agent-based model, and the regression model represented by vector autoregressive model [2,3]. In practical application, such methods can get more detailed results from the actual economic operation data, but whether the actual effect of the policy can be truly separated from the data changes, and the degree of coherence between theoretical results and reality is easy to be questioned. The second category is to evaluate the rationality and feasibility of the policy content itself, which mainly includes qualitative evaluation methods represented by fuzzy comprehensive evaluation and policy literature measurement methods gradually emerging in recent years. The policy text is transformed into structured data that can be calculated, and the policy document and its implementation process and effect are systematically and comprehensively investigated and analyzed. Then a reasonable evaluation conclusion is drawn. This paper mainly reviews the second type of methods.

At present, the evaluation of policy content itself is mostly carried out by the scoring method. Susana et al. [4] applied Delphi method and analytic hierarchy process to build an innovation policy evaluation index system. The financial macro network method proposed by Stolbova et al. [5] can effectively supplement the shortcomings of existing climate policy assessment methods. Martinho et al. [6] proved through research that literature metrology, literature review and quantitative methods can better explore the dimensions related to soil legislation and policy in the context of the European Union. Helming et al. [7] conducted an empirical analysis by establishing a general equilibrium model. Nag et al. [8] divided policies into three categories of urgent, mandatory and general policies from the perspective of policy objectives and characteristics, and made corresponding evaluation indicators. Abotah et al. [9] evaluated and selected existing policies through a hierarchical decision-making model. BEN-NETT [10] developed the logical framework of policy evaluation in the field of public administration that consider factors in multiple stages from resource input to final goal output. Compared with traditional policy evaluation methods, PMC index model [11] is a relatively advanced policy quantitative assessment way in the world. Kuang et al. [12] established a PMC index model of cultivated land conservation policy, also compared it with other indexes to show the curve characteristics of this kind of policy, so as to evaluate it objectively. In recent years, the comprehensive scoring method based on PMC index has been tentatively applied to the quantitative assessment of innovation policy in science and technology [13], carbon emission reduction policy [14], insurance policy [15], etc., showing good applicability to this kind of research. Although this method has a more general quantitative assessment of policies, similar to other subjective evaluation methods, it is highly subjective in evaluation variable assignment and scoring, and the results are easy to be questioned. At present, the commonly used policy evaluation methods at home and abroad mainly include rooted theory, entropy weight

TOPSIS method, LDA subject model, differential differential model, PMC index model and so on. Among them, the advantages of PMC index model are mainly manifested in three aspects: (1) The research object focuses on the policy document itself, which can not only conduct quantitative evaluation of a single policy, but also analyze the consistency of multiple policies; (2) The setting of variables at all levels of the model refers to the results of text mining, effectively avoiding the arbitrariness of subjective evaluation; (3) Abundant evaluation indicators can be used to analyze the internal heterogeneity and pros and cons of policies in multiple dimensions, and the policy evaluation results can be visually displayed through PMC curved graph, which is convenient to find the weak points of policies.

Policy text analysis method is a kind of policy analysis method that have emerged in recent years, including qualitative analysis represented by interpretive analysis, discourse analysis and comparative analysis of policy text and quantitative analysis represented by content analysis, quantitative analysis and content mining of policy text. Based on co-word analysis and cluster analysis methods, scholars Huang et al. [16] analyzed the change of thematic key point of the science and technology innovation policy and found that such policies have undergone significant policy thematic changes in the four fields of " international collaboration ", " manpower resource", "mechanism reform" and "research and progress priorities". Scholars Tan et al. [17] used social network analysis, semantic network analysis, correspondence analysis and other methods to analyze the differences and similarities of high-frequency words in these documents of the five-year plan for coal industry from 1996 to 2016, and obtained the changing trend of the direction and focus of such policies. Although this kind of policy text analysis method is still in its infancy, it enriches the data sources of policy analysis and is more objective and fairer than the traditional qualitative research of policy interpretation or subjective scoring methods.

## 2.2 Research on sports related polices

Accompanied by improvement of the status of sport and sports industry in national development, many scholars have carried out research on sports related policies, focusing on mass sports, school sports and competitive sports [18]. Wang [19] and Zhang [20], taking competitive sports practice as the direction, pointed out that the development characteristics of competitive sports policy are stay in step with the overall development of sport in China. Li and Guo [21] pointed out that the development of school physical education policy in China is characterized by the change of policy value orientation from the national standard to the human standard, and from the emphasis on decision-making guidance to the emphasis on science. Hu and Tang [22] analyzed the changes of the sports industry from 1993 to 2015, and concluded that the evolution direction of the sports industry was to strengthen the leading role of sports industry, improve market environment, and enhance the legal consciousness. They found that in the formulation of sport policy, three aspects should be considered: decision-making, implementation and evaluation. Wang et al. [23] elaborated the formulation, objectives, implementation agencies, measures and achievements of China's sports industry policy through in-depth exploration. Lin [24] found through statistical data research that China's sports industry has different levels of policy advantage tilt and advanced development. These indicators reflect the government's preference in resource allocation and can be used to measure the effect of policies.

To sum up, there are still some breakthroughs in the existing research on policy evaluation. From the content level, the focus on the special issue of sports industry policy evaluation is insufficient, and the lack of research on policy content makes most of the existing research only applicable to post-evaluation. From the method level, the existing evaluation model has

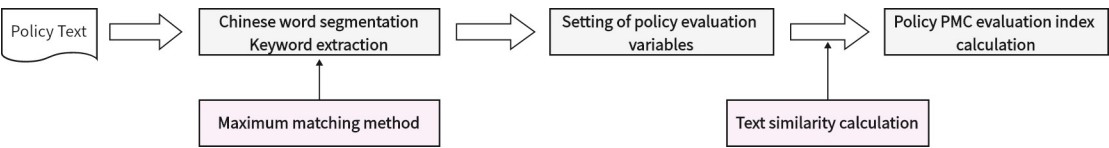

**Fig 2. PMC index model construction and variable assignment method flow of policy evaluation.**

some problems such as lack of comprehensiveness in factor selection, strong subjectivity in valuation and low precision. Therefore, this study constructs a PMC index model, calculates index assignment based on intelligent text similarity algorithm, and makes quantitative evaluation on the existing representative sports industry policy content. It aims to provide reference for policy makers to analyze and adjust existing policies or to issue new policies. The technology roadmap is shown in Fig 2.

## 3. Research methods

This study selects the related policies of sports industry as the object, and uses a method of combining the analysis of policy evolution characteristics, policy text mining analysis and index calculation to carry out the study on quantitative evaluation of sports industry policy. Among them, the characteristics of policy evolution are obtained by analyzing and summarizing the rules of sports industry policy evolution, and analysis of text mining includes high-frequency word statistics. PMC index model is proposed by Estrada based on the "Omnia Mobilis" hypothesis. As a quantitative policy evaluation and analysis method, PMC index model is prevailingly employed to evaluate policies through consistency evaluation indicators, so that decision makers can identify the consistency, advantages and disadvantages of policies. There are four steps including variable classification and parameter setting, setting up multi-input output table, calculating PMC index and drawing PMC surface map.

### 3.1 PMC index analysis

The PMC index claims that any relevant variables should be taken into account, so the variables should be selected as widely as possible without ignoring any possible relevant variable. We can understand the internal consistency of a policy from each dimension, and its surface formed by measuring the PMC index can intuitively reflect the situation of each dimension of the policy. It also can conduct multidimensional evaluation of the policy effectiveness and conduct specific analysis of single indicators.

First of all, evaluation variables are determined and parameters are set. For the sake of evaluating China's sports industry policy in a more targeted manner, variables in this study are set according to of reference to the policy assessment indicators put forward by Estrada [11], united with the results of sports industry policy text content mining, the first and second indexes are determined.

Secondly, the multi-input-output table is constructed. As a data analysis tool, this table can reserve a mass of data, measure and evaluate various indicator variables multidimensional, and reflect the developing procedure of a specific policy. The table is composed of a number of first-level indicators and unrestricted second-level indicators, the weight of every indicator is equal and independent of each other. According to formula, the secondary variables are assigned and put into the input-output table. The score of the first-level indicators is calculated on the basis of Formula ③. The values of the first-level indicators are the ratio of the total score of the second-level indicators to the amount of second-level indicators.

Finally, the PMC index is calculated and the surface is constructed. Finally, the PMC index of the sports industry policy is calculated on the basis of the formula, that is, total up all the scores of first-level indicators of one policy.

$$X \sim [0, 1] \tag{①}$$

$$X \sim \{PR : [0, 1]\} \tag{②}$$

$$X_i = \sum_{j=1}^{n} \frac{X_{ij}}{n(X_{ij})} \tag{③}$$

$$\text{PMC index} = \sum_{j=1}^{m} X_i = \sum_{j=1}^{m} \left( \sum_{j=1}^{n} \frac{X_{ij}}{n(X_{ij})} \right) \tag{④}$$

Where i represents the first-level indicators, j represents the second-level indicators, Xij represents a second-level indicators, m represents the amount of first-level variables, and n or n (Xij) represents the amount of second-level variables under a first-level variable. According to Estrada [11], the PMC-Index could be divided into four levels of consistency: inferior (0–3.99), acceptable (4.00–6.5), great (6.51–7.50) and excellent (7.51–9.00), which provides basic criteria for comparing different policies. The PMC surface represents the value of PMC index in the form of stereoscopic images, which can intuitively display the advantages and disadvantages of sports industry policies. PMC surfaces are always dependent on the matrices. The PM matrix is a matrix of 3 rows and 3 columns, and the PMC matrix corresponding to the PMC surface is calculated by the Formula ⑤.

$$\text{PMC} - \text{Surface} = \begin{bmatrix} X_1 & X_4 & X_7 \\ X_2 & X_5 & X_8 \\ X_3 & X_6 & X_9 \end{bmatrix} \tag{⑤}$$

## 3.2 Index assignment based on text similarity algorithm

A policy often involves multiple policy objectives or policy tools, and it is easy to arbitrariness to judge the topic solely through text reading or score from the title. Therefore, text mining is applied to the index assignment for analysis. Establish policy database V(d) for sports industry-related policies, where the text information in each policy text pi is $w_i(d)$. First of all, it is necessary to conduct word segmentation processing for the policy text, and adopt a more mature maximum matching method to match the maximum symbol string of the text with the dictionary entry. If it fails to match, a Chinese character will be cut off and continue to match until the corresponding word is found in the dictionary. According to TF-IDF method [25,26], set $TF_i$ as word frequency, N as total number of documents, and frequency of text appearing in document set as n, then:

$$TF - IDF(w_k) = w_k = \frac{tf_k * \log(N/n_k)}{\sqrt{\sum_{k=1}^{n} tf_k^2 * [\log(c + N/n_k)]^2}} \tag{⑥}$$

The traditional spatial vector model for text similarity calculation uses the Angle cosine of the policy text feature vector to represent the distance:

$$Sim(p_1, p_2) = \frac{\sum_{k=1}^{n} w_{1k} * w_{2k}}{\sqrt{\sum_{k=1}^{n} w_{1k}^2 * \sum_{k=1}^{n} w_{2k}^2}} \tag{⑦}$$

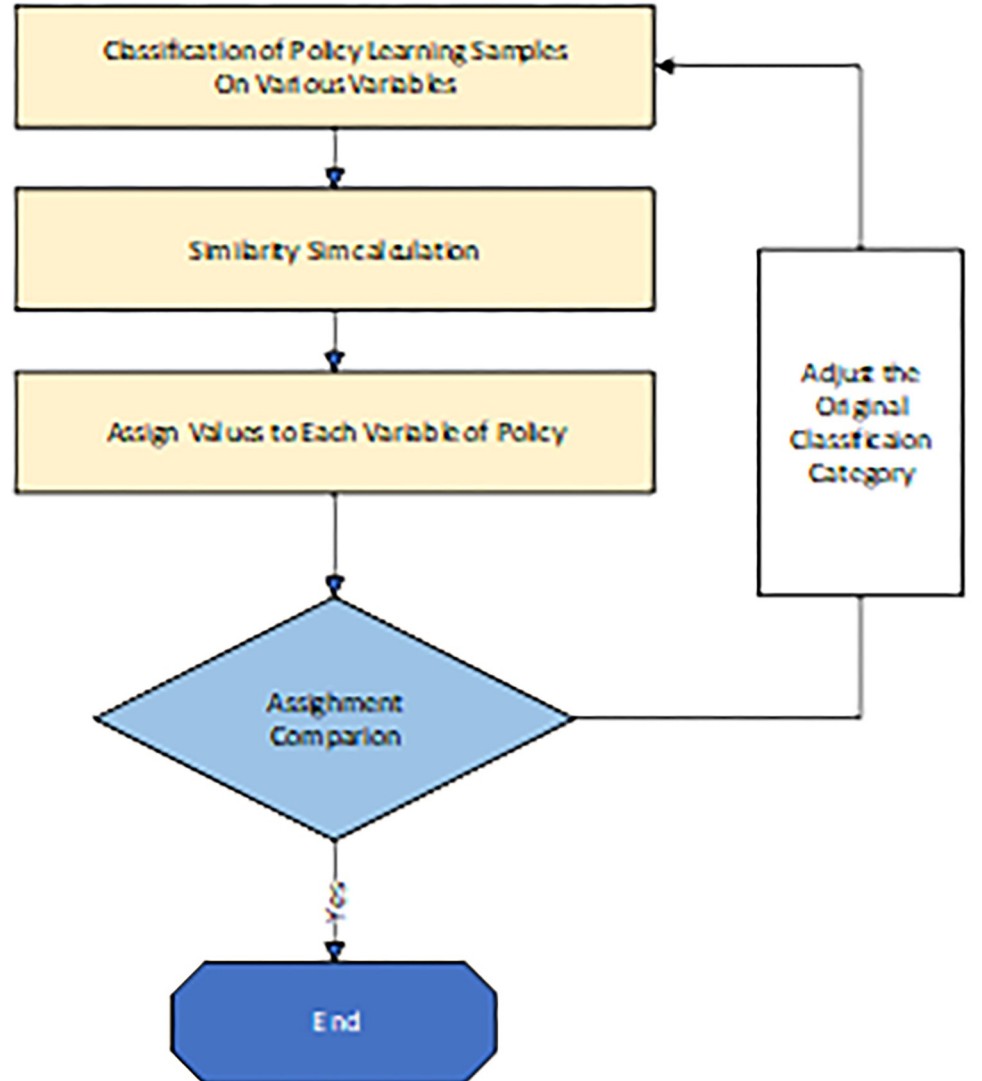

**Fig 3. Process of constructing learning samples for policy assessment.**

When assigning values to quantitative evaluation variables, the Formula ⑦ is used to classify the documents, and W represents the word frequency of each word in the word frequency vector. According to the policy text corresponding to each variable, that is, the learning sample, through the intelligent calculation method, if a new policy text is similar to it, it is considered that the new policy is assigned a value of 1 on the evaluation variable; otherwise, it is 0. However, in the actual classification process, there are no mature classification samples for reference for sports industry policy research. Consequently, this paper adopts the learning sample classification method. [27], as shown in Fig 3. Aiming at the classification accuracy of learning samples to be 100%, theoretical classification is first given, and part of the learning samples are predictive classified. If it is inconsistent with the theoretical classification, the original classification is adjusted, and the learning sample for subsequent similarity calculation is finally obtained through repeated process. Given a threshold, if the similarity is greater than the threshold, the indicator is assigned a value of 1.

## 4. Results and analysis

For the sake of ensuring the completeness and accuracy of the selected policies, this paper is based on the catalog of current and effective sports laws, regulations, regulations, normative documents and institutional documents issued by the General Administration of Sport of China, and adopts a method of "Internet search + cross-check" to search the website of the Chinese government, the official website of the General Administration of Sport of China, the official website of the Ministry of Education, and the official websites of provincial governments. With the keywords "sports industry", "mass sports" and "school sports", etc., the policy documents were searched, and priority was given to the policy documents with higher efficacy level and sports industry theme words in the title. Through manual screening, we filtered out duplicate documents as well as nonoperational policy documents such as meeting minutes, congratulatory messages, and bulletins. Finally, a total of 611 policy documents were collected.

### 4.1 Analysis of the evolution of sports polices in China

Since the reform and opening policy was promulgated, Chinese government has shifted its focus to economic construction, and the sports industry has also been on the "fast track" of development in line with the requirements of The Times. According to the number and focus of China's sports industry policies, this study takes the evolution history of sports industry in China as basis, and takes the year of 1978 as the starting point. Evolution and change process of sports industry policy is divided into the four phases: germination period (1978–1992), active exploration period (1993–2000), rapid development period (2001–2013), and comprehensive development period (2014-present).

1. Germination Period (1978–1992)
   Since the year of 1978, China's economy started to get a rapid expansion. In 1979, China regained its legal identity in the International Olympic Committee, and the industrialization of China's sport market gradually took shape. The reform of school sport, mass sport, competition training, sport science and technology was fully launched, and relevant policies are constantly seeking directions and actively exploring, trying to make sport realize industrial development. In 1984, State Physical Culture and Sports Commission of China issued the "Notice on the Further Development of Sport", which clearly proposed the increase of sport funds and infrastructure construction, and included it in the national economic plan, and the "Decision on economic System Reform" promulgated by the government of China in the same year also made the sporting goods manufacturing industry get the initial bud. The sportswear industry represented by Putian in Fujian province emerged. In 1986, the State Sport Commission issued the "Decision on the reform of the sport system", pointing out that the use of "multiple management" methods to strengthen the construction of sport venues, and began to take the lead in the reform of individual sport. The introduction of a few policies cannot form a joint force and break the "institutional cage" that restricts the evolution of the sports industry, but under active exploration, industrialization attribute of the evolution of sport is gradually highlighted. In 1992, in response to the "Decision to accelerate the development of the Tertiary Industry" promulgated by the government of China, the then National Sport Commission formally proposed the concept of sports industry. At this point, the conception of sports industry was taken by degrees, but in this period, sports industry only serves as a way to promote economic development, and its position in economic development has not been paid enough attention.

2. Active Exploration Period (1993–2000)
   As the last ten years before the 21st century, the market economy had gradually stabilized.

The focus of the relevant sports industry policies formulated by the state is mainly on the establishment and improvement of related laws and regulations system. China started to strengthen standardized management of the sports industry. In 1993, the "Opinions on Deepening the reform of Sport" was issued, and it marked the official beginning of the reformation of China's sport system. This document also clearly proposed the goal of the development of sports industrialization, which is mainly guided by macro policies. In the year of 1995, "The Outline for the Development of the Sports industry (1995–2010)" was issued, and it clarified the composition of sports industry. In the same year, the first law regulating the high-quality development of sport came into being, bringing the evolution of sport in China into a new stage that governing sport according to law. Subsequently, various policies on the sports industry have been promulgated. At this stage, the status of the sports industry in economic development was established, and the policy content is more targeted. With the coordination of resource allocation and government tax support, the sports industry policy system started scaling.

3. Rapid Development Period (2001–2013)

At the early 21st century, China's sports industry ushered in new development opportunity as it enters the new century. In July 2001, China successfully got the opportunity to host the Olympic Games, and in November of the same year, China successfully joined the WTO. In this context, China's sports consumption continues to be active, the sport market system continues to improve, the evolution of the sports industry was highly valued by government departments of different levels, and sports industry policy also followed the current, seizing the opportunities of The Times. The biggest feature of the policy at this stage is that the main body of the policy gradually changed from the unitary main body dominated by the former State Sport Commission to the diversified main body dominated by departments at all levels. The State's emphasis and planning on the sports industry provided strong support and target for the development of the sports industry. The General Office of the State Council issued the "Guiding Opinions on Accelerating the Development of the Sports industry" in 2010, and the General Administration of Sport issued the "12th Five-Year Plan for the sports industry" in 2011, Chinese government issued special policies and special plans at the national level for the first time to support the evolution of the sports industry. At this stage, the central departments at all levels, mainly the General Office of the State Council, issued significantly more policies than the previous stage, through further market supervision, tax incentives and other aspects of support, the sports industry ushered into a rapid development period.

4. Comprehensive Development Period (2014-present)

Since the year of 2014, the national level to promote the evolution of sports industry policy documents frequently. In the year of 2014, the State Council officially issued the "Several opinions on Accelerating the development of sports industry to promote sports consumption", and promoted the evolution of sports industry to the state strategy level, China's sports industry development ushered into the "second spring", into an overall development period. "Guiding opinions of the general office of the state council on accelerating the development of the fitness and leisure Industry" in 2016 and "Guiding opinions of the general office of the state council on accelerating the development of the sport competition and performance industry" in 2018 had positive impact on the evolution of China's sports industry. At this stage, the number, subject and impact of the polices are much higher than the previous stages, and the policy content is also more plentiful, and it began to pay attention to the evolution of emerging industries such as sport tourism and exhibition, and promotes the radiation scope of China's sports industry to be broader. Under the influence of

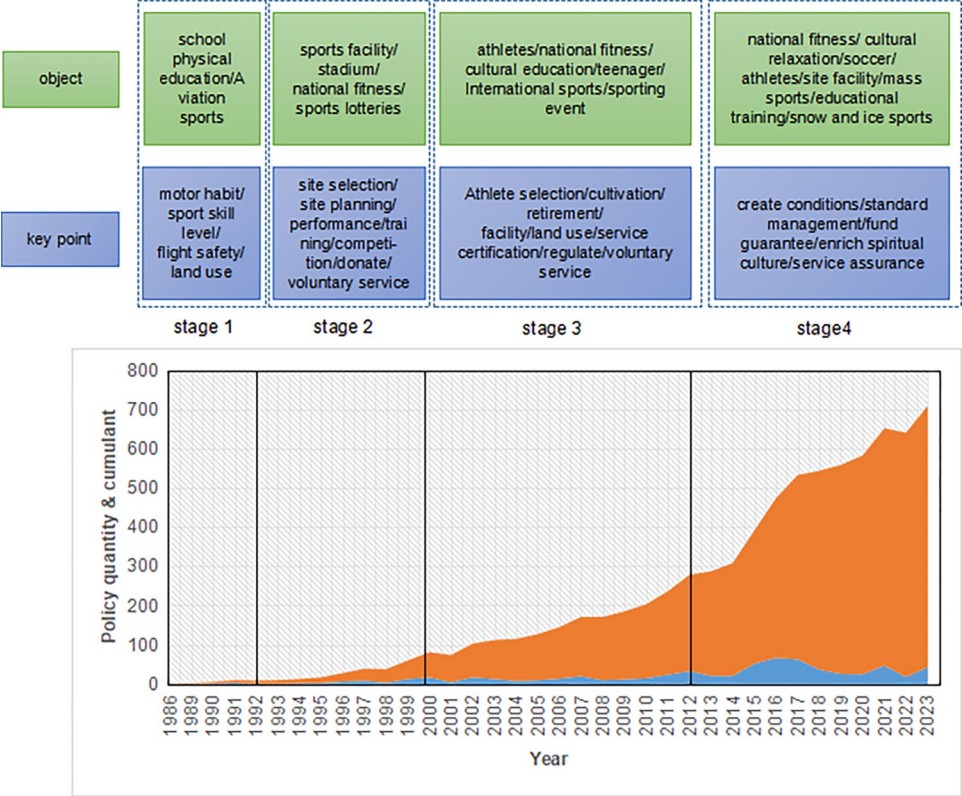

**Fig 4. Policy stage evolution diagram.**

the policy dividend, the evolution of China's sports industry repeatedly achieved good results, the total amount increased year by year, and the status in the national economy also continued to improve, in the year of 2018, China's sports industry was more than 1% of GDP for the first time. With the continuous optimization of policies, it is necessary to accelerate the development and improve the level of the sports service industry through openness. Now, due to policy incentives, a large number of top foreign clubs come to China every year to explore the market. From large to strong and forming global competitiveness is the inevitable path for industrial development.

The sport policy stage evolution is shown in Fig 4.

## 4.2 PMC index analysis

**4.2.1 Text mining and policy selection.** The policy text of this study includes both central government policies and local government policies. Since provincial governments in China are subject to the leadership and restriction of the central and national government [28], this political relationship not only ensures the consolidated implementation of major strategies of the government, but also stimulates the initiative and sense of responsibility of local governments. Therefore, this paper selects 230 high-frequency words in central and national policies as the basis for constructing quantitative evaluation variables of policies. The central and national texts were imported into ROST software for word frequency statistics, and general service words and adverbs that were not significant to this field were screened and filtered. In this paper, based on relevant literature research and expert opinions, high-frequency words are screened. The higher the word frequency, the more important the word was.

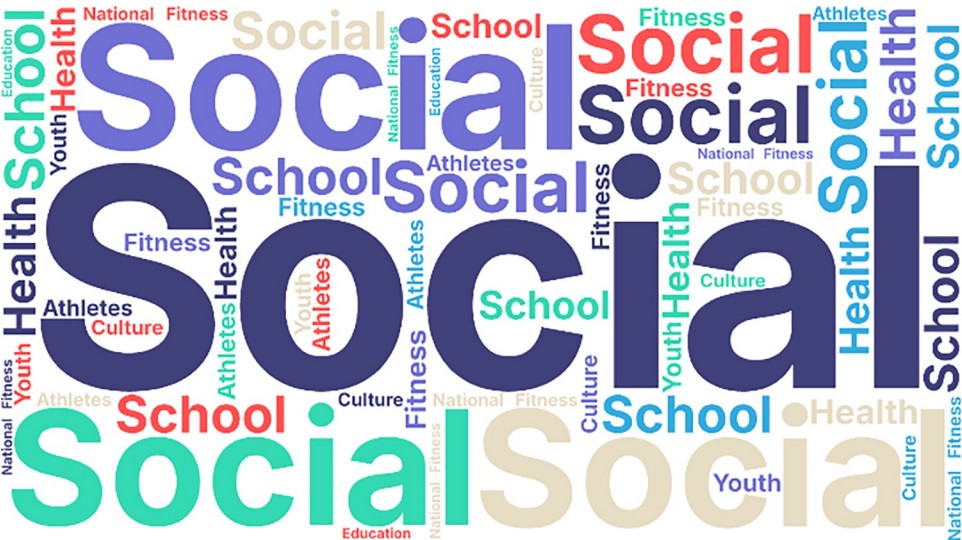

**Fig 5. Word cloud of some high-frequency words.**

As can be seen from Fig 5, high-frequency words like "School", "Athletes", "Youth" and "Seniors" appearing, shows that the development of the sports industry has a positive impact on people of different ages. The appearance of high frequency words such as society and school show that with the idea of lifelong sport, social sports into the school has become the core concern of government. The emergence of high-frequency words such as coaches, training, and sport venues shows that government focus more on the supporting conditions for the evolution of the sports industry.

On the basis of referring to the existing PMC policy evaluation model, this study modified some evaluation dimensions according to the high-frequency words analysis results, we finally established 9 first-level indicators and 47 second-level indicators of the PMC index model, which is shown in Table 1.

Finally, this study selects 6 policies that at the national level that cover a wide range of areas, have a large time span and involve more areas. The selected indicators are more suitable for evaluating comprehensive policies, and some adjustments should be made to specific policies. The reason for choosing these policies is that P1 has for the first time elevated nationwide fitness as a state strategy, which is an important document leading the evolution of China's sports industry in the course of comprehensively building moderately prosperous society and deepening the reform comprehensively. P2 is an important policy jointly issued by 12 departments, which defines the goals and tasks, main work, and investment guarantee measures of the elderly sport work under the new situation, is a practical measure for the functional departments of the national government to promote establishing a social system for healthy elderly care, and is an important guiding document for the elderly sport work in the new era. A leading sporting nation is the goal and task of the reform and development of China's sport work in the new period, P3 is the embodiment of implementing the spirit of the national important conference, and it is to further clarify the goals, tasks and measures for the construction of sports power, further defines the goals, tasks and measures of the construction of sport power. P4 is a guiding document for the implementation of the important discourse on education, it has made an overall design for the reform, development and high-quality construction of school aesthetic education under the new background of times. P5 is a policy document issued by the State Council specifically for the first time that for the construction of fitness facilities

**Table 1. Setting of policy quantitative evaluation variables.**

| first-level variable | second-level variable | second-level variable evaluation standard | variable basis |
|---|---|---|---|
| **(X1) Nature of the policy** | (X1:1) forecast | Whether involves forecasting | Ruiz Estrada [11] |
| | (X1:2) recommendation | Whether involves recommendations | |
| | (X1:3) regulation | Whether involves regulation, 1 for yes, | |
| | (X1:4) Guidance | Whether involves guidance | |
| | (X1:5) Description | whether involves the description of the development status | |
| | (X1:6) Diagnostic | Whether involves a summary of existing problems | |
| **(X2) Policy duration** | (X2:1) Long-term | Whether covers more than 5 years | Ruiz Estrada [11] |
| | (X2:2) Medium-term | Whether covers the content of 3–5 years | |
| | (X2:3) Short-term | Whether involves the content within 3 years | |
| **(X3) Policy level** | (X3:1) National level | Whether involves the national level | Dong et al. [29]; High-frequency word |
| | (X3:2) Provincial level | Whether involves the provincial level | |
| | (X3:3) Prefecture level | Whether involves prefecture level | |
| **(X4) Policy instruments** | (X4:1) Supply-oriented | Whether involves government support in terms of production factors such as technology, facilities and capital | Fang Liu et al. [30] High-frequency word statistics |
| | (X4:2) Demand-oriented | Whether involves direct support in stimulating demand, such as organizing events, attracting social participation, exchanges and cooperation | |
| | (X4:3) Environmental | Whether involves improving the rule of law, optimizing industrial layout and other methods to provide a good environment for industrial development | |
| **(X5) Policy area** | (X5:1) Economic | Whether involves the economic area | Kuang et al. [12]; High-frequency word statistics |
| | (X5:2) Social | Whether involves the social field | |
| | (X5:3) Law | Whether involves the law field | |
| | (X5:4) Science and technology | Whether involves the field of science and technology | |
| | (X5:5) Cultural | Whether involves the cultural field | |
| | (X5:6) Exchange | Whether involves the field of exchange | |
| **(X6) Policy function** | (X6:1) Industrial development | Whether involves industrial development | Laibing Lu et al. [31] |
| | (X6:2) Market development | Whether involves market development | |
| | (X6:3) Protect people's livelihood | Whether involves protecting people's livelihood | |
| | (X6:4) Specification guidance | Whether involves specification guidance | |
| | (X6:5) Government functions | Whether involves government functions | |
| | (X6:6) System constraint | Whether involves system constraint | |
| **(X7) Policy audience** | (X7:1) Government departments | Whether involves government departments | High-frequency word statistics |
| | (X7:2) Enterprise organization | Whether involves enterprise organization, | |
| | (X7:3) Social public | Whether involves the public, | |
| | (X7:4) School | Whether involves schools at all levels | |
| **(X8) Policy focus** | (X8:1) Overall organization | Whether involves strengthening the organization | High-frequency word statistics |
| | (X8:2) Capital investment | Whether involves the capital investment | |
| | (X8:3) Science and technology innovation | Whether involves the science and technology innovation | |
| | (X8:4) Team building | Whether involves the content of team building | |
| | (X8:5) Security development | Whether involves the content of security development | |
| | (X8:6) Specific object | Whether involves the content of a specific object | |
| | (X8:7) Event | Whether involves the content of event | |
| | (X8:8) Sports consumption | Whether involves the content of sports consumption | |
| | (X8:9) Feature project | Whether involves the content of the feature project | |
| | (X8:10) Property rights protection | Whether involves the content of property rights protection | |
| | (X8:11) Infrastructure | Whether involves the content of infrastructure | |
| | (X8:12 Industrial land | Whether involves the content of industrial land | |

*(Continued)*

**Table 1.** (Continued)

| first-level variable | second-level variable | second-level variable evaluation standard | variable basis |
|---|---|---|---|
| (X9) Policy Approach | (X9:1) Mandatory | Whether involves the mandatory content | Kuang et. Al [12] |
| | (X9:2) Incentive | Whether involves incentive content | |
| | (X9:3) Service | Whether involves the service content | |
| | (X9:4) marketability | Whether involves marketability content | |

Note: All the values in the Table 2 are taken as 1 for yes, 0 for no.

and mass sport, which fully reflects the great importance to developing mass sport and improving the standard of public services for nationwide fitness. P6 summarizes the achievements of China's sports industry during the "13th Five-Year Plan" period, systematically plans top-level planning document for the evolution of China's sports industry during the "14th Five-Year Plan" period, and makes a comprehensive deployment for the evolution for the next five years, which is an important policy document in this period. The list of policies is shown in Table 2.

After variable parameters are determined, text similarity algorithm is used to assign values to the input and output matrix. Other central and national policies other than the above six policies are selected to establish a policy database. Based on word segmentation, maximum matching method is adopted to classify the policy texts. According to the corresponding policy texts of each variable, that is, learning samples, if a new policy text is similar to it, the threshold is set at 0.8 after testing, and if the similarity is greater than 0.8, the variable is assigned as 1. The multi-input and output matrix is obtained, as shown in Table 4. Eq (4) was applied to calculate the PMC-Index value of each strategy successively. Results are shown in Tables 3 and 4.

For visualization to present the differences and disadvantages of sports industry policies, we apply the Origin software to curve the PMC surface map, shown as Figs 6–11 In these graphs, the numbers 1,2, and 3 on the left matches the X-axis of the matrix, and the numbers 1,2, and 3 on the right matches the Y-axis of the matrix.

In addition, concave and convex of the surface indicates different PMC index values. The more convex of the surface shows that the index value corresponding to the policy is higher, and the more concave of the surface shows that the index value corresponding to the policy is lower. Three dimensions correspond to the PMC matrix.

**Table 2. List of typical policies.**

| Number | Policy Name | Release year |
|---|---|---|
| P1 | Several Opinions on Accelerating the development of sports industry and promoting sports consumption | 2014 |
| P2 | Notice on the issuance of "Opinions on Further Strengthening Physical Education for the Elderly under the New Situation" | 2015 |
| P3 | Notice on printing and distributing the outline of building a strong sport country | 2019 |
| P4 | "Opinions on comprehensively strengthening and improving school physical education in the new era" and "Opinions on comprehensively strengthening and improving aesthetic education in schools in the new era" | 2020 |
| P5 | Opinions on Strengthening the construction of National Fitness venues and facilities to Develop Mass sport | 2021 |
| P6 | The 14th Five-Year Plan for sport Development | 2021 |

**Table 3. Multi-input-output table for 6 sports industry policies.**

| Primary Variables | Secondary Variables | P1 | P2 | P3 | P4 | P5 | P6 |
|---|---|---|---|---|---|---|---|
| X1 | X1:1 | 0 | 0 | 1 | 0 | 0 | 0 |
|    | X1:2 | 1 | 1 | 1 | 1 | 0 | 1 |
|    | X1:3 | 1 | 0 | 1 | 0 | 1 | 0 |
|    | X1:4 | 1 | 1 | 1 | 1 | 1 | 1 |
|    | X1:5 | 1 | 0 | 0 | 0 | 0 | 1 |
|    | X1:6 | 0 | 0 | 0 | 0 | 0 | 1 |
| X2 | X2:1 | 1 | 1 | 1 | 1 | 1 | 1 |
|    | X2:2 | 0 | 0 | 0 | 0 | 0 | 0 |
|    | X2:3 | 0 | 0 | 0 | 0 | 0 | 0 |
| X3 | X3:1 | 1 | 1 | 1 | 1 | 1 | 1 |
|    | X3:2 | 0 | 0 | 0 | 0 | 0 | 0 |
|    | X3:3 | 0 | 0 | 0 | 0 | 0 | 0 |
| X4 | X4:1 | 1 | 1 | 1 | 1 | 1 | 1 |
|    | X4:2 | 1 | 1 | 1 | 1 | 1 | 1 |
|    | X4:3 | 1 | 1 | 1 | 1 | 1 | 1 |
| X5 | X5:1 | 1 | 1 | 1 | 1 | 0 | 1 |
|    | X5:2 | 1 | 1 | 1 | 1 | 1 | 1 |
|    | X5:3 | 0 | 0 | 1 | 1 | 1 | 1 |
|    | X5:4 | 1 | 0 | 1 | 0 | 0 | 1 |
|    | X5:5 | 1 | 1 | 1 | 1 | 0 | 1 |
|    | X5:6 | 0 | 0 | 1 | 0 | 0 | 1 |
| X6 | X6:1 | 1 | 0 | 1 | 0 | 0 | 1 |
|    | X6:2 | 1 | 1 | 1 | 0 | 0 | 1 |
|    | X6:3 | 1 | 1 | 1 | 1 | 1 | 1 |
|    | X6:4 | 1 | 1 | 1 | 1 | 1 | 1 |
|    | X6:5 | 1 | 1 | 1 | 1 | 1 | 1 |
|    | X6:6 | 1 | 0 | 0 | 1 | 0 | 1 |
| X7 | X7:1 | 1 | 1 | 1 | 1 | 1 | 1 |
|    | X7:2 | 1 | 0 | 1 | 0 | 0 | 1 |
|    | X7:3 | 1 | 1 | 1 | 0 | 1 | 1 |
|    | X7:4 | 1 | 0 | 1 | 1 | 1 | 1 |
| X8 | X8:1 | 1 | 1 | 1 | 1 | 1 | 1 |
|    | X8:2 | 1 | 1 | 1 | 1 | 1 | 1 |
|    | X8:3 | 1 | 1 | 1 | 0 | 0 | 1 |
|    | X8:4 | 1 | 1 | 1 | 1 | 1 | 1 |
|    | X8:5 | 0 | 0 | 0 | 1 | 1 | 1 |
|    | X8:6 | 0 | 1 | 0 | 1 | 1 | 1 |
|    | X8:7 | 1 | 0 | 1 | 1 | 1 | 1 |
|    | X8:8 | 1 | 1 | 1 | 0 | 0 | 1 |
|    | X8:9 | 0 | 0 | 0 | 0 | 0 | 1 |
|    | X8:10 | 1 | 0 | 0 | 0 | 0 | 1 |
|    | X8:11 | 1 | 1 | 1 | 1 | 1 | 1 |
|    | X8:12 | 1 | 0 | 1 | 0 | 1 | 1 |
| X9 | X9:1 | 1 | 0 | 0 | 1 | 1 | 0 |
|    | X9:2 | 1 | 1 | 1 | 1 | 1 | 1 |
|    | X9:3 | 1 | 1 | 1 | 1 | 1 | 1 |
|    | X9:4 | 1 | 0 | 1 | 0 | 1 | 1 |

**Table 4. PMC Index evaluation of six policies.**

|  | P1 | P2 | P3 | P4 | P5 | P6 | 均值 |
|---|---|---|---|---|---|---|---|
| X1 | 0.67 | 0.33 | 0.67 | 0.33 | 0.33 | 0.67 | 0.5 |
| X2 | 0.33 | 0.33 | 0.33 | 0.33 | 0.33 | 0.33 | 0.33 |
| X3 | 0.67 | 1 | 0.67 | 0.33 | 0.67 | 1 | 0.72 |
| X4 | 1 | 1 | 1 | 1 | 1 | 1 | 1 |
| X5 | 0.67 | 0.5 | 1 | 0.67 | 0.33 | 1 | 0.70 |
| X6 | 1 | 0.67 | 0.83 | 0.67 | 0.5 | 1 | 0.78 |
| X7 | 1 | 0.5 | 1 | 0.6 | 0.75 | 1 | 0.81 |
| X8 | 0.75 | 0.58 | 0.58 | 0.66 | 0.67 | 1 | 0.71 |
| X9 | 1 | 0.5 | 1 | 0.6 | 1 | 1 | 0.85 |
| PMC index | 7.09 | 5.41 | 7.08 | 5.19 | 5.58 | 8 | 6.39 |
| Level | Good | Acceptable | Good | Acceptable | Acceptable | Excellent | |

### 4.2.2 Typical policy analysis

From the calculation results and efficiency, it can be seen that the policy features are more abundant than the traditional tool-industrial value chain two-dimension analysis framework. From the title and theme, we can see that the types of the six selected policies are different. The rank of PMC-Index value is as follows: P6 > P1 > P3 >P5 > P2 > P4, in which policy P6 is excellent. High-quality policies can promote the evolution of the sports industry, improve industrial structure and people's well-being. According to the current policy evaluation results, the structure and direction of China's sports industry policies are generally reasonable. To be specific, the amount of sports industry policies in China is growing rapidly recently, reflecting

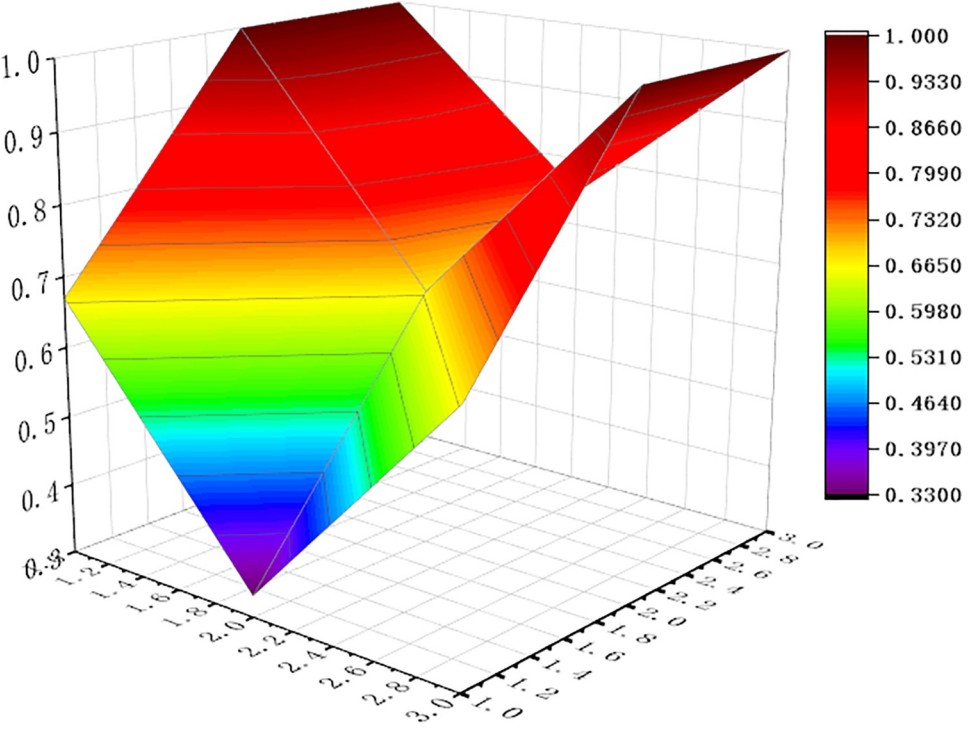

**Fig 6. PMC surface of P1.**

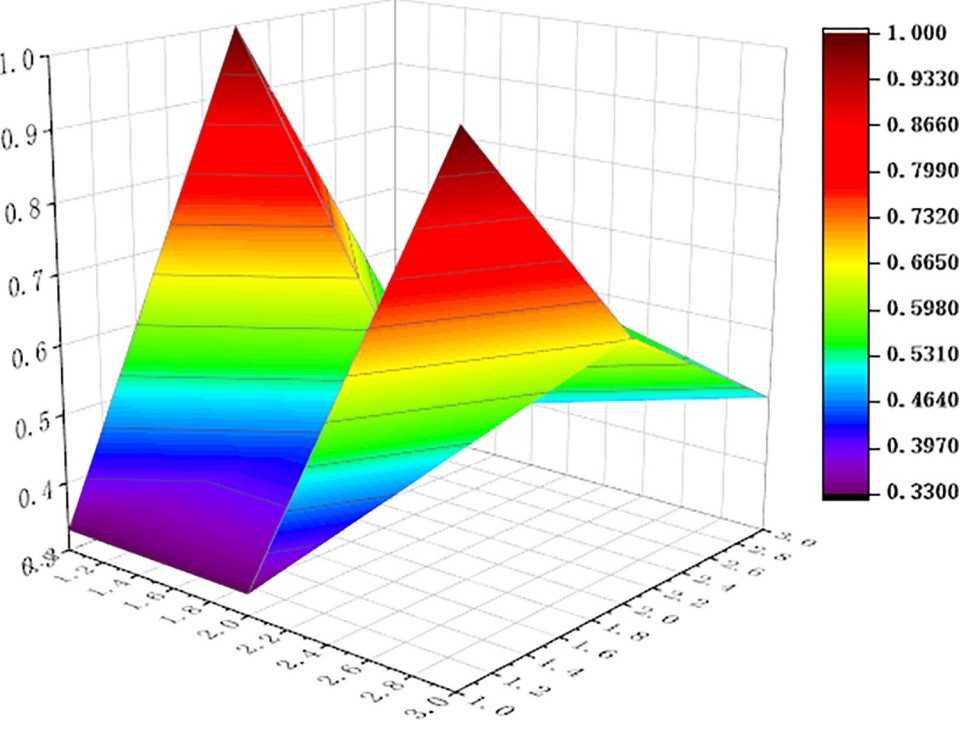

**Fig 7. PMC surface of P2.**

the unprecedented level of national importance attached to sport, which will have a far-reaching impact. At the present stage, the typical policies in terms of policy tools, domain, function, audiences and methods have achieved the role of accelerating the construction of China's sport power, accelerating the integrated development of the industry and encouraging the innovation of new quality and productivity, and can provide support for the establishment of a sports industry framework with scientific and reasonable layout, fully function and complete classification. According to the 2021 PwC Sports Industry Survey Report released in December 2021, the global sports industry has had a compound annual growth rate of 4.9% over the past 3–5 years, and is expected to maintain a growth rate of 4.9% over the next 3–5 years; The sports industry in North America is expected to maintain a strong growth trend in market size over the next 3–5 years; Meanwhile, against the backdrop of technological progress and policy support, the sports industry in Asia, Africa, and South America has enormous development potential. Therefore, Chinese government needs to work harder on improving its sports industry policy. In addition, the average value of policy timeliness (X2) is low, indicating that the six policies do not fully take into account the significance of multi-long-term strategies and short-term goals.

Two typical policies, P6 and P1, were selected for analysis: (1) The PMC index value of policy P6 was 9, ranking first among the six policies. It can be seen from Fig 8 that, except for the policy timeliness (X2), the other surfaces of policy P6 are smooth, and it means that policy P6 has defects in terms of policy timeliness and policy level. This is because the issuing agency of policy P6 is the General Administration of Sport of China, and the formulation of the plan is based on the functions of the General Administration of Sport, without multi-subject interaction. This policy combines the economic and social changes and the development status of China's sports industry, and makes effective design arrangements in terms of policy audiences,

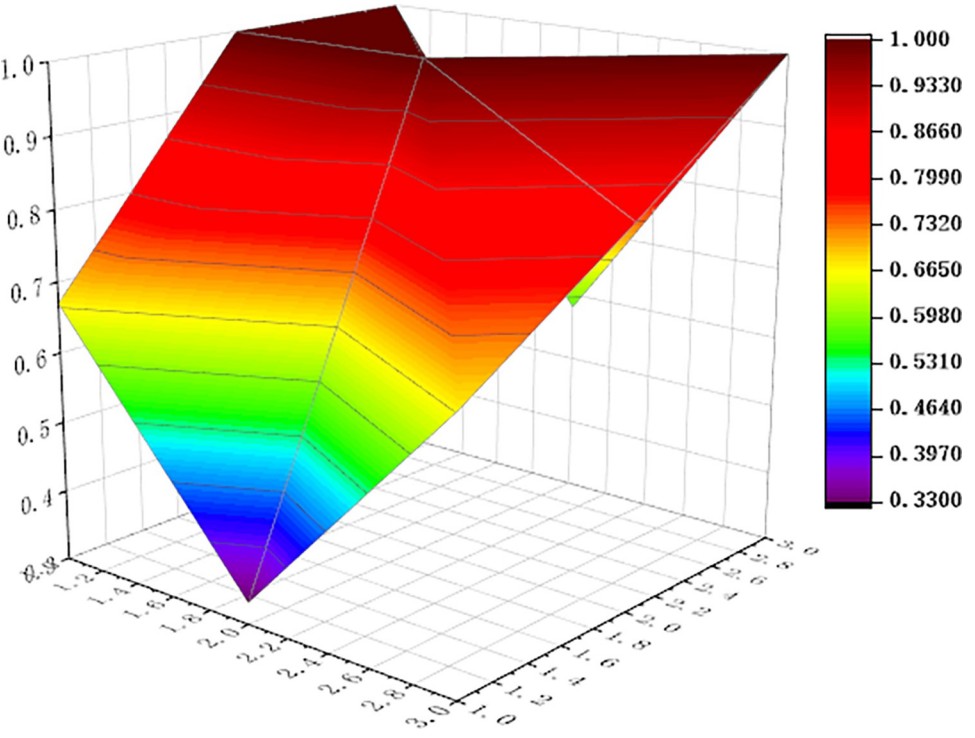

**Fig 8. PMC surface of P3.**

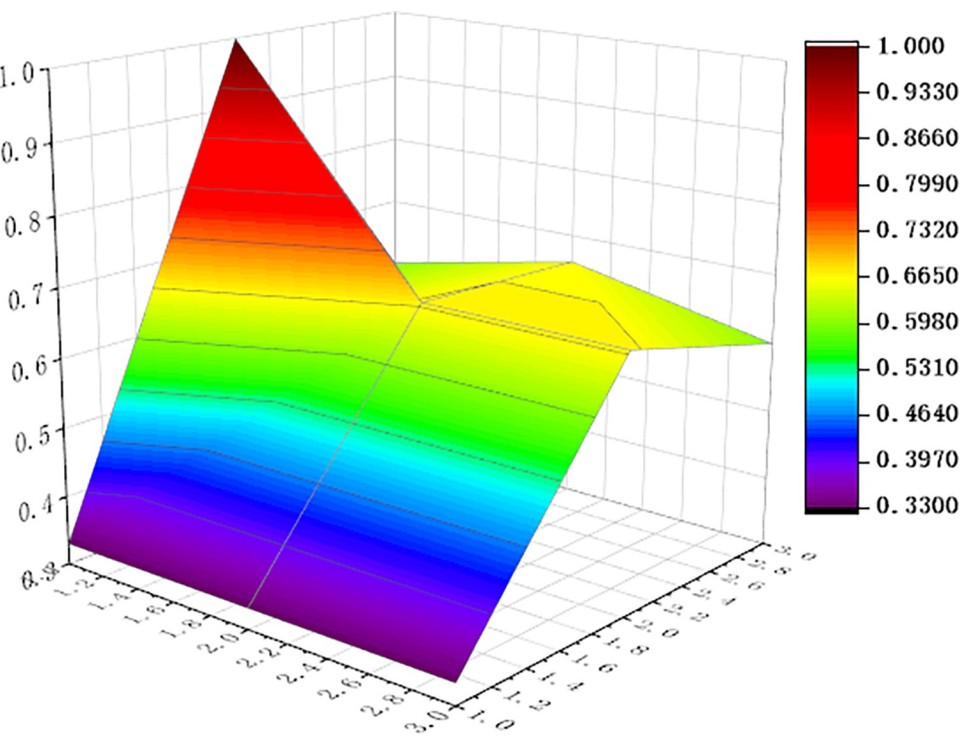

**Fig 9. PMC surface of P4.**

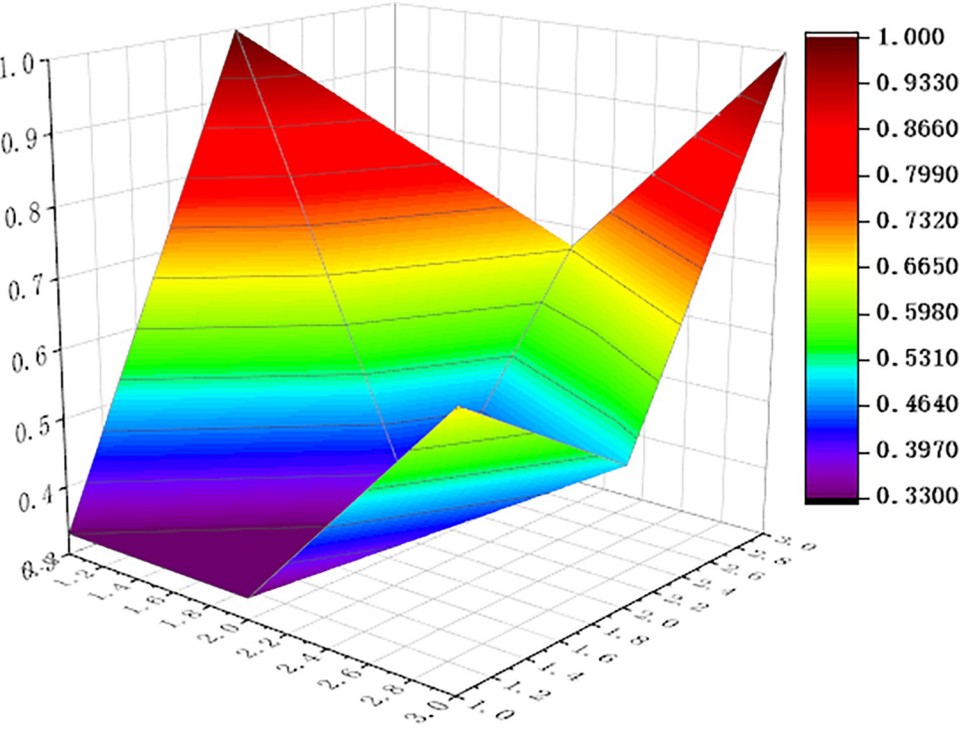

**Fig 10. PMC surface of P5.**

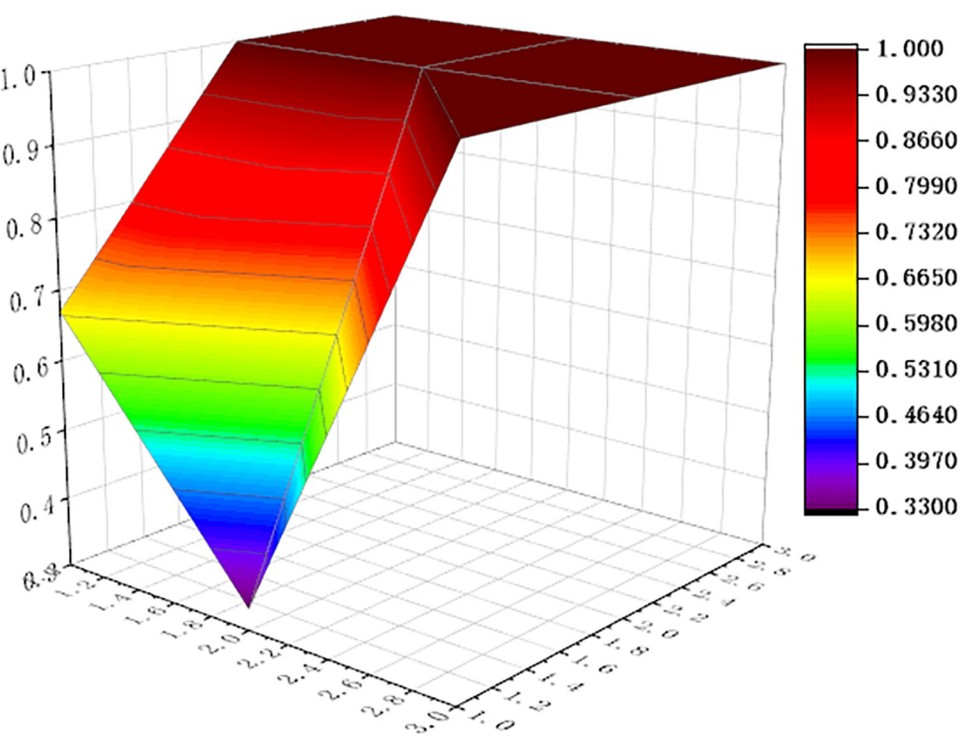

**Fig 11. PMC surface of P6.**

priorities, methods, areas, functions, tools, etc. It is a relatively scientific and feasible policy. (2) The PMC index value of policy P1 is 7.09, ranking second among the six policies. The P1 policy is designed to promote the sports industry as a significant power for China's economic restructuring and upgradation. As can be seen in Fig 4, (X4) policy tools, (X6) policy functions, (X7) policy audiences, and (X9) policy methods are significantly convex, with a darker color and a higher value of 1.00. It shows that policy P1 has the features that of diversified tools, comprehensive functions and wide audience, and considers the combination of ways to promote the implementation of policies. However, similar to policy P6, policy timeliness (X2) and policy level (X3) showed a significant downturn. The reason is that P1 policy is relatively macro and the critical part of the policy is single, and it lacks consideration of how to coordinate the implementation of the policy content among various departments at various stages, and it needs to support the specific implementation of the policy.

**4.2.3 Evaluation dimension analysis.** Nature of the policy (X1). The nature of policy refers to the role of guidance, specification and prediction. The mean value of the six typical policies of China's sports industry is 0.5, which indicates that China's sports industry policies have certain capabilities of supervision, suggestion and prediction. Among them, except for policy P1, P3 and P6, the others whose index values are less than 0.5. This difference shows that there are great differences in the nature of China's sports industry policies, of which half of the sports industry policies are mainly suggestions, supervision and guidance, and lack of diagnosis of problems at the present stage and prediction of future development. The main problems caused by the lack of policy analysis on the current situation include out-of-control policy implementation, additional implementation, etc. The problems caused by the lack of government policy forecast mainly include policy lag, low efficiency of resource allocation, and the inconsistency between social costs and benefits. Effective analysis of current problems and forecast of future development can further improve the pertinence of policies.

Policy duration (X2). The average duration of the six typical sports industry policies is 0.33. Specific analysis shows that the current policy time of the sports industry is mainly long-term planning and medium and long-term planning, lacking short and medium term planning and guidance. Reasonable and moderate policy objectives and timeliness planning and timely adjustment are the prerequisite to achieve the expected effect of the policy, and we should keep a watchful eye on the organic combination of varying duration. First of all, it is necessary to formulate a clear long-term goal, and then formulate a plan for a period of time in accordance with the principle of "near thin, far thick", and then adjust and modify the future plan according to the effect of the implementation of the plan and changes in the internal and external environment. Meanwhile it is essential to be concerned about the degree of accomplishment of policy objectives within the time limit, as well as the supervision and evaluation of implementation effects.

Policy level (X3). The research results show that the average level of the six typical sports industry policies is 0.72. Specific analysis shows that the typical sports industry policies mainly provide detailed planning and policy recommendations from the national level, and cooperate with the deployment of provincial and municipal work. As a macro-policy at the national level, it needs to be further supported by multi-level specific policies to promote its implementation, and has achieved the expected policy effect.

Policy Instrument (X4). The research results display that the mean value of the six typical sports industry policy tools is 1, which is the highest dimension of PMC mean value, indicating that China has a high diversity of policy tools, and these six policies comprehensively consider all kinds of policy tools, which is beneficial to enhance the effectiveness of policies.

Policy area (X5). The research results display that the mean value of the six typical sports industry policies is 0.7, indicating that the field distribution of the six policies is relatively

good. However, the values of all policies are lower than the mean, except for P3 and P6. This indicates that some sport policies have not fully taken sport as an important part of social culture in the formulation process, especially the external and cultural aspects have not considered the influence of the sports industry, and the development of sport is also subject to such external factors.

Policy function (X6). Policy function refers to the social role that can be played after policy formulation and implementation. In this study, the policy functions mainly include industrial development, market development, and ensuring people's livelihood. The average function of the six policy strategies of China's sports industry policy is 0.78, indicating that China's sports industry policy has better functions and is an effective and feasible policy. It is worth noting that the values of policies P2, P4 and P5 are all less than 0.78, indicating that these policies need to be improved in terms of policy functions.

Policy audience (X7). The research results show that the average audience of the six typical sports industry policies is 0.81, indicating that the direction of the strategy is clear. Among them, the policy object value of policy P1, policy p3 and policy p6 is 1.00, much higher than 0.81, and the PMC index of these 6 strategies is higher, indicating that the wider the range of objects targeted by the strategies, the higher the evaluation value. Through the research, it is found that strengthening the coordination between various government departments is an important means to promote the evolution of the industry. Most policies mention that the government can promote the development of the sports industry by deepening the influence of policies and increasing investment. Enterprises are the "main battlefield" of China's sports industry development, financial support, tax incentives, research and development support for enterprises is an important support of China's sports industry policy to enterprises, but in the research process found that most of the policies are too macro, did not introduce specific help measures. Social organization is the source of the evolution of sports industry in our country, and to promote the decoupling of sport social organization and government, "social sport" is both a trend and a trend, but in the text of sports industry policy for social organization is relatively scarce. The individual is the ultimate landing point and the ultimate beneficiary of the evolution of sports industry. Although at present, China's sports industry policy pays more and more attention to the "human", there are fewer policy provisions aimed at the individual in the policy text. Therefore, it is essential to strengthen the policy ratio between individuals and social organizations.

Policy focus (X8). The research results reveal that the mean value of the six typical sports industry policies is 0.71, except for P1 and P6, the rest of the policies are lower than the average, among which there are fewer policies involving characteristic projects and property rights protection, and insufficient support for the evolution of water sport, e-sport and other projects. With the enhancement of national fitness awareness, for all kinds of projects should take certain targeted guidance, encouragement and support measures to promote the evolution of sports industry. The application of intellectual property protection in China's sports industry policy is rare. In recent years, China's various industries have paid more and more attention to the protection of intellectual property rights of enterprises and individuals. If the protection is not properly maintained, piracy may be rampant, which will cause enterprises and individuals in the sports industry to lose the enthusiasm for investment, research and development and production, and ultimately affect the integrated evolution of the sports industry. Therefore, in the formulation and implementation of China's sports industry policy, we should appropriately tilt to the direction of intellectual property protection.

Policy Approach (X9). The research results show that the average of the six typical sports industry policies is 0.85, and the policies are more reasonable and diversified, mainly through guiding, encouraging and supporting the main body of the sports industry to promote the

development of the sports industry, and the use of mandatory and market policies are relatively few, which will lead to the poor standardization of the industry and the loss of development vitality. It leads to the reduction of self-restraint and development ability, so China's sports industry policy should reasonably use "comprehensive measures" according to the actual situation. As far as possible, China's sports industry should adapt itself to the market, so as to reduce the government's active intervention.

There are clear industrial development scale goals, as well as specific measures such as land planning and facility support in P1. For the first time, it has been clarified that the sports industry plays an important role in promoting economic growth. P1 officially kicked off the prelude to accelerating the development of sports industry and promoting sports consumption. P5, based on the new situation, new tasks, and new requirements faced by China's sports development in the future, focuses on the new development stage, implements the new development concept, constructs a new development pattern, and draws a grand blueprint for comprehensively building a sports power and promoting high-quality sports development. Therefore, these two policies have higher scores.

In recent years, programmatic documents such as the "Outline for the Construction of a powerful sports Country" and the "Fourteenth Five-Year Plan for the Development of the sports Industry" have been issued, pointing out the direction for the deepening development of the sports industry and putting forward specific development goals and measures. In January 2023, the newly revised Sports Law of the People's Republic of China set up a special chapter on "Sports industry", further clarifying the functional value of the sports industry from the legal level. Under the promotion of a series of policies, China's sports industry has achieved remarkable results. On the one hand, the government actively supports the development of sports enterprises by increasing financial investment and optimizing the business environment. For example, through the establishment of sports industry investment funds, provide loans and other ways to help sports enterprises to solve financing problems; Optimize the business environment of the sports industry by simplifying the approval process and lowering the market entry threshold; On the other hand, the government also creates market opportunities for the development of the sports industry and consolidates the basic conditions for the development of the sports industry by holding large-scale sports events and constructing sports infrastructure. The implementation of these policies and measures has not only promoted the rapid development of the sports industry, but also injected new impetus into economic and social development.

## 5. Conclusions and suggestions

Recently, the research of policy evaluation methods and models has been widely concerned, and the evaluation of policy text content is one of the ways. On the basis of traditional policy evaluation research, this study introduces an intelligent index assignment method. The index assignment based on text similarity algorithm can realize automatic PMC index assignment for policy text through sample training, which can enhance the objectivity of policy evaluation and improve computational efficiency and accuracy. In this study, we first sort out the sports industry policies and find that the number of sports industry policies is increasing year by year. On the one hand, the formulation of policies is continuously detailed in key areas, and on the other hand, it is continuously extended horizontally in the industrial value chain. In order to avoid the arbitrariness of variable subjective scoring, a spatial vector model is used to assign variables according to the results of policy text mining. The results show that this method can evaluate policies more comprehensively than the traditional multidimensional analysis method. Among the six typical policies selected, differences in the content, function and scope

of each policy lead to different policy evaluation scores. Policy P6 has a more comprehensive effect and relatively higher score. This method shows strong applicability in policy evaluation. According to the evaluation results, this paper puts forward the following suggestions:

1. Sports industry policy should take into account the direction of guidance and the formulation of rules. Policies such as planning and opinions have played a strategic guiding part in the evolution of the sports industry and the promotion of local government work, while detailed policies such as opinions on promoting and regulating the development of social sport clubs have played an operational role in the actual industry development promotion work, so both are very important in the policy mix. The development of the sports industry has reached a certain scale. In the face of periodic problems in the development process such as unbalanced development and incoordination in various fields of sport, these key problems and new problems should be specifically written into the plan, and practical measures and detailed rules should be made to accelerate the healthy evolution of the industry.

2. Strengthen the top-level design of sports industry policies, improve the policy supervision and accountability system, and establish a practical feedback mechanism. The formulation, implementation, and feedback of sports industry policies is a complex task, and the top-level design of the policy system should carry out medium—and long-term overall deployment and systematic planning for the development of the sports industry from the national level. We should adhere to the coordination and unity of policy system design, and form a structurally reasonable, hierarchically clear, and scientifically complete policy system.

3. The effectiveness level, evolution process, and market matching degree of sports industry policies directly affect the size of policy effectiveness, and improving policy coordination and optimizing policy structure are guarantees for enhancing policy implementation effectiveness. To this end, it is necessary to first strengthen communication and coordination among various government departments, enhance the synergy among policy makers at different levels, and ensure the rationality and systematicity of various policy formulations.

In subsequent research, the variable settings of the PMC index model should be appropriately improved based on the actual situation of various levels of sports industry policies. The first level variable settings should strive for greater universality, and the second level variable settings should strive for breadth to strengthen the scientific and accurate evaluation of policies.

## Supporting information

**S1 File. Sample data: Article minimum data set.**
(RAR)

## Author Contributions

**Conceptualization:** Liang Ma.

**Investigation:** Liang Ma.

**Methodology:** Liang Ma.

**Supervision:** Zhaoyang Pan.

**Validation:** Zhaoyang Pan.

**Visualization:** Zhaoyang Pan.

**Writing – original draft:** Zhaoyang Pan.

**Writing – review & editing:** Liang Ma.

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
