## [Decision Letter · Decision Letter 0]

15 Oct 2024

PONE-D-24-43160Research on evolution and evaluation of Chinese sports industry based on intelligent analysisPLOS ONE

Dear Dr. Ma,

Thank you for submitting your manuscript to PLOS ONE. After careful consideration, we feel that it has merit but does not fully meet PLOS ONE’s publication criteria as it currently stands. Therefore, we invite you to submit a revised version of the manuscript that addresses the points raised during the review process.

We look forward to receiving your revised manuscript.

Kind regards,

Tinggui Chen

Academic Editor

PLOS ONE

Journal Requirements:

Additional Editor Comments:

I have completed my evaluation of your manuscript. The reviewers recommend reconsideration of your manuscript following major revision. I invite you to resubmit your manuscript after addressing the comments below.

Reviewers' comments:

Reviewer's Responses to Questions

**Comments to the Author**

1. Is the manuscript technically sound, and do the data support the conclusions?

Reviewer #1: Yes

Reviewer #2: Yes

2. Has the statistical analysis been performed appropriately and rigorously? 

Reviewer #1: Yes

Reviewer #2: Yes

3. Have the authors made all data underlying the findings in their manuscript fully available?

Reviewer #1: Yes

Reviewer #2: Yes

4. Is the manuscript presented in an intelligible fashion and written in standard English?

Reviewer #1: Yes

Reviewer #2: Yes

5. Review Comments to the Author

Reviewer #1: The manuscript titled "Research on evolution and evaluation of Chinese sports industry based on intelligent analysis" (PONE-D-24-43160), need to be revised properly before resubmission:

1. The manuscript lacks a clear statement of the specific research objective. While the importance of evaluating Chinese sports policies is emphasized, it would be helpful to explicitly outline the research question. For instance, clarifying whether the study is focused on evaluating the effectiveness, evolution, or strategic value of sports policies would improve the focus.

2. The use of the PMC index for policy evaluation is innovative, but the explanation of the methodology is somewhat unclear. The process for selecting high-frequency words and mapping them to indicators needs more clarity. What criteria were used to ensure that the high-frequency words directly reflect policy effectiveness? The reliance on text mining may not capture policy intent fully.

3. The paper mentions using 611 policy documents, but the criteria for selecting specific policies for analysis is not thoroughly explained. Were these policies chosen based on their scope, impact, or frequency of mention? Additionally, there is no explanation of why only six policies were used in the final analysis. Addressing this gap is crucial for replicability.

4. The division of policy evolution into four phases is informative, but the rationale behind the time periods selected (1978–1992, 1993–2000, etc.) is not well substantiated. Are these divisions based on significant historical events or policy shifts? Providing more justification for these divisions would strengthen the historical analysis.

5. The manuscript mentions that China’s sports policy framework lags behind that of Europe and the USA. However, there is no detailed comparison provided. Including a comparative analysis of China’s sports policies with global standards (e.g., EU or USA policies) would provide greater context and show where China excels or falls short.

6. The paper heavily relies on the text mining approach to analyze policy documents, but this could miss out on nuances such as policy implications or effectiveness, which are often implicit. Text mining alone may not capture the full depth of policy documents, and supplementary qualitative analysis could provide a more robust evaluation.

7. While the PMC index evaluates the content of policies, there is no discussion on how these policies have been implemented or their real-world outcomes. Including a section on the actual impacts of these policies (e.g., increases in sports participation, economic growth, etc.) would provide a more comprehensive view of policy effectiveness.

8. The study claims that Chinese sports policies are scientifically sound and effective. However, this generalization may not hold for all policy areas. For instance, specific areas like mass sports or competitive sports may face unique challenges that are not covered by the broad evaluation of the policy framework. A more segmented analysis based on policy types could enhance the findings.

9. The paper uses PMC index values to evaluate policy effectiveness, but there is no mention of statistical validation. Are these results statistically significant?

Reviewer #2: 1. Title & Abstract

• Title: The title clearly reflects the scope of the research, but it could be more concise. Consider: “Intelligent Analysis of the Evolution and Evaluation of China’s Sports Industry.”

• Abstract: The abstract provides a good overview of the study, but it is somewhat dense. To enhance clarity, it might help to break up longer sentences and focus on key findings. For example, more emphasis could be placed on the practical implications of the study and what “new directions” were found.

2. Introduction

• The introduction effectively sets the context by highlighting the importance of the sports industry in China. However, more attention could be given to explaining why the sports industry needs intelligent analysis. For a more compelling start, you might include specific data showing the sports industry’s impact on China’s GDP or other socioeconomic metrics to justify the need for optimization.

• It would also help to briefly introduce the PMC index and text mining earlier to give readers a sense of the novel methodology used.

3. Literature Review

• The literature review is comprehensive, with well-cited references to policy assessment methods and related research on sports policy. However, the discussion around methods could be better structured by summarizing the strengths and weaknesses of each method at the end of the review. This would clearly show how the chosen PMC index and text mining are appropriate for the study.

• Some sections in the review (e.g., on European Union policies) feel slightly disconnected from the Chinese context. Consider focusing more directly on Chinese studies or global comparisons relevant to China’s sports industry to strengthen relevance.

4. Methodology

• PMC Index: The explanation of the PMC index and its use in evaluating policy effectiveness is well-done, though it is technical. You might include a simpler explanation or visual to help readers unfamiliar with the model.

• Text Mining: This section could benefit from a more detailed explanation of how text mining was applied. Did the authors use specific software for this analysis? Including a step-by-step breakdown (alongside Figure 1) of how the text mining fed into the PMC index could be valuable.

• Data Collection: The method of collecting policy documents is solid, but you could clarify the selection process a bit more. For example, why were 611 policy documents chosen, and what criteria were used for inclusion/exclusion?

5. Results and Analysis

• The results section provides an insightful analysis of the evolution of China’s sports policies. However, this section is data-heavy and would benefit from more concise summary tables and visuals to break up the text.

• The analysis of the four developmental phases of the sports industry (Germination, Active Exploration, etc.) is strong, but it could use more comparative insights—perhaps by linking China’s progress with international examples.

• In the typical policy analysis, a more qualitative discussion of why policies like P6 or P1 performed better would enrich the quantitative analysis.

6. Discussion

• The discussion around policy instruments, timeliness, and level of policy focus is detailed and insightful. However, this section could be made more engaging by linking findings back to the original research question and emphasizing the implications for future policy-making.

• The limitations of the study are not clearly addressed. A paragraph on potential biases in text mining, or limitations due to the PMC model, could strengthen the transparency of the study.

7. Conclusion and Recommendations

• The conclusion is strong in terms of summarizing the results, but it could be more concise. The recommendations are practical and well-grounded in the analysis.

• It might help to emphasize the future applicability of this intelligent analysis in other sectors or for other countries. For instance, how could this PMC model benefit other developing economies’ sports industries?

8. Clarity & Language

• Overall, the language is clear, but some sections are overly technical. Simplifying jargon-heavy sentences, especially in the methodology and results sections, would help make the paper accessible to a broader audience.

• A few minor grammatical errors were noted, particularly with article usage and sentence length. Proofreading would help with flow.

9. Figures and Tables

• The figures (like the PMC surface maps) are useful but could be better labeled to guide interpretation. Adding more explanatory captions would help.

• The inclusion of high-frequency words in Table 1 is insightful, though this could be integrated into a more visual format (e.g., word clouds) to make it more engaging.

Final Thoughts:

This is a strong paper with a solid methodology, but it would benefit from streamlining certain sections and emphasizing the real-world implications of the findings. The application of intelligent analysis and the PMC index is innovative, but the value it brings to policy-making could be more directly emphasized throughout the paper.

6. PLOS authors have the option to publish the peer review history of their article (what does this mean?). If published, this will include your full peer review and any attached files.

Reviewer #1: No

Reviewer #2: No

---

## [Author Response · Author response to Decision Letter 0]

2 Dec 2024

Additional Editor Comments:

I have completed my evaluation of your manuscript. The reviewers recommend reconsideration of your manuscript following major revision. I invite you to resubmit your manuscript after addressing the comments below.

Reviewers' comments:

Reviewer's Responses to Questions

Comments to the Author

1. Is the manuscript technically sound, and do the data support the conclusions?

Reviewer #1: Yes

Reviewer #2: Yes

2. Has the statistical analysis been performed appropriately and rigorously?

Reviewer #1: Yes

Reviewer #2: Yes

3. Have the authors made all data underlying the findings in their manuscript fully available?

Reviewer #1: Yes

Reviewer #2: Yes

4. Is the manuscript presented in an intelligible fashion and written in standard English?

Reviewer #1: Yes

Reviewer #2: Yes

5. Review Comments to the Author

Reviewer #1: The manuscript titled "Research on evolution and evaluation of Chinese sports industry based on intelligent analysis" (PONE-D-24-43160), need to be revised properly before resubmission:

1. The manuscript lacks a clear statement of the specific research objective. While the importance of evaluating Chinese sports policies is emphasized, it would be helpful to explicitly outline the research question. For instance, clarifying whether the study is focused on evaluating the effectiveness, evolution, or strategic value of sports policies would improve the focus.

Accept. In the introduction, it is added to explain the significance of the number of policies and text analysis on the development of sports industry.

2. The use of the PMC index for policy evaluation is innovative, but the explanation of the methodology is somewhat unclear. The process for selecting high-frequency words and mapping them to indicators needs more clarity. What criteria were used to ensure that the high-frequency words directly reflect policy effectiveness? The reliance on text mining may not capture policy intent fully.

Accept. On the basis of text mining, literature reading and expert opinion are used to ensure that the included words are representative. A supplementary explanation is given in the part of methodology.

3. The paper mentions using 611 policy documents, but the criteria for selecting specific policies for analysis is not thoroughly explained. Were these policies chosen based on their scope, impact, or frequency of mention? Additionally, there is no explanation of why only six policies were used in the final analysis. Addressing this gap is crucial for replicability.

Accept. Added explanation in Section 4.2.1. In the analysis of policy selection, with the level of policy formulation, coverage time, coverage area and other aspects as the main consideration, the comprehensive policy at the national level is selected with a wider coverage, a larger time span, and more areas involved.

4. The division of policy evolution into four phases is informative, but the rationale behind the time periods selected (1978–1992, 1993–2000, etc.) is not well substantiated. Are these divisions based on significant historical events or policy shifts? Providing more justification for these divisions would strengthen the historical analysis.

Accept. Add a description in Section 4.1. 1992, the Chinese government officially put forward the concept of sports industry. The year 2000 is the first year of the new century and China successfully applied for the right to host the Olympics. According to the growth of the number of policies, the growth rate of the number of policies has accelerated significantly after 2014. According to the above reasons, the industrial development is divided into four stages

5. The manuscript mentions that China’s sports policy framework lags behind that of Europe and the USA. However, there is no detailed comparison provided. Including a comparative analysis of China’s sports policies with global standards (e.g., EU or USA policies) would provide greater context and show where China excels or falls short.

Accept. Quoting authoritative reports to argue and explain in Section 4.1.2. According to the 2021 PwC Sports Industry Survey Report released in December 2021, the development of the sports industry in North America is better than that in Asian countries, and the development of the sports industry in Asian countries requires strong support from technology and policies.

6. The paper heavily relies on the text mining approach to analyze policy documents, but this could miss out on nuances such as policy implications or effectiveness, which are often implicit. Text mining alone may not capture the full depth of policy documents, and supplementary qualitative analysis could provide a more robust evaluation.

Accept. In section 4.2.2, based on quantitative evaluation, a qualitative interpretation of the evaluated policy text is conducted to deeply analyze the connotation and significance of the policy.

7. While the PMC index evaluates the content of policies, there is no discussion on how these policies have been implemented or their real-world outcomes. Including a section on the actual impacts of these policies (e.g., increases in sports participation, economic growth, etc.) would provide a more comprehensive view of policy effectiveness.

Accept. Add the explanation and analysis in the sector of introduction and 4.2.3 to analyze the impact of policies on the economy and society.

8. The study claims that Chinese sports policies are scientifically sound and effective. However, this generalization may not hold for all policy areas. For instance, specific areas like mass sports or competitive sports may face unique challenges that are not covered by the broad evaluation of the policy framework. A more segmented analysis based on policy types could enhance the findings.

Accept. Section 4.2.1 adds a description of the applicability of the model, which is more suitable for comprehensive policy evaluation. For policy evaluation in specific fields, the model needs to be adjusted.

9. The paper uses PMC index values to evaluate policy effectiveness, but there is no mention of statistical validation. Are these results statistically significant?

Accept. The conclusions obtained by PMC evaluation and analysis method are mainly qualitative conclusions, which have certain statistical significance for large-scale policy evaluation. However, due to the limitation of the length of the paper, statistical analysis is not carried out in this paper.

Reviewer #2: 1. Title & Abstract

• Title: The title clearly reflects the scope of the research, but it could be more concise. Consider: “Intelligent Analysis of the Evolution and Evaluation of China’s Sports Industry.”

• Abstract: The abstract provides a good overview of the study, but it is somewhat dense. To enhance clarity, it might help to break up longer sentences and focus on key findings. For example, more emphasis could be placed on the practical implications of the study and what “new directions” were found.

Accept. However, this study focuses on policy analysis, so the term ‘policy’ is still retained in the title

2. Introduction

• The introduction effectively sets the context by highlighting the importance of the sports industry in China. However, more attention could be given to explaining why the sports industry needs intelligent analysis. For a more compelling start, you might include specific data showing the sports industry’s impact on China’s GDP or other socioeconomic metrics to justify the need for optimization.

• It would also help to briefly introduce the PMC index and text mining earlier to give readers a sense of the novel methodology used.

Accept. The concepts of PMC index and text mining are added in the introduction, and the general introduction is made in the research methods part.

3. Literature Review

• The literature review is comprehensive, with well-cited references to policy assessment methods and related research on sports policy. However, the discussion around methods could be better structured by summarizing the strengths and weaknesses of each method at the end of the review. This would clearly show how the chosen PMC index and text mining are appropriate for the study.

• Some sections in the review (e.g., on European Union policies) feel slightly disconnected from the Chinese context. Consider focusing more directly on Chinese studies or global comparisons relevant to China’s sports industry to strengthen relevance.

Accept. Add related content about PMC index and text mining in the literature review section.

Delete the parts in the review that disconnected from the Chinese context.

4. Methodology

• PMC Index: The explanation of the PMC index and its use in evaluating policy effectiveness is well-done, though it is technical. You might include a simpler explanation or visual to help readers unfamiliar with the model.

• Text Mining: This section could benefit from a more detailed explanation of how text mining was applied. Did the authors use specific software for this analysis? Including a step-by-step breakdown (alongside Figure 1) of how the text mining fed into the PMC index could be valuable.

• Data Collection: The method of collecting policy documents is solid, but you could clarify the selection process a bit more. For example, why were 611 policy documents chosen, and what criteria were used for inclusion/exclusion?

Accept. The intuitive display of PMC index is mainly in matrix form, which visualizes the matrix in three dimensions.

We use the ROSTCM6 text software to assist in text mining and analysis in our research, which is explained in section 4.2.1

Add the policy screening principles in the chapter 4.

5. Results and Analysis

• The results section provides an insightful analysis of the evolution of China’s sports policies. However, this section is data-heavy and would benefit from more concise summary tables and visuals to break up the text.

• The analysis of the four developmental phases of the sports industry (Germination, Active Exploration, etc.) is strong, but it could use more comparative insights—perhaps by linking China’s progress with international examples.

• In the typical policy analysis, a more qualitative discussion of why policies like P6 or P1 performed better would enrich the quantitative analysis.

Accept. Simplified Tables 2 and 3

Add relevant content in section 4.1

Add relevant content in section 4.2.2

6. Discussion

• The discussion around policy instruments, timeliness, and level of policy focus is detailed and insightful. However, this section could be made more engaging by linking findings back to the original research question and emphasizing the implications for future policy-making.

• The limitations of the study are not clearly addressed. A paragraph on potential biases in text mining, or limitations due to the PMC model, could strengthen the transparency of the study.

Accept. Add a discussion on problem orientation in the conclusion section.

Add descriptions of research deficiencies in the conclusion section.

7. Conclusion and Recommendations

• The conclusion is strong in terms of summarizing the results, but it could be more concise. The recommendations are practical and well-grounded in the analysis.

• It might help to emphasize the future applicability of this intelligent analysis in other sectors or for other countries. For instance, how could this PMC model benefit other developing economies’ sports industries?

Accept. Change the conclusion to make it more grounded.

The laws of industrial development have commonalities and provide similar insights for the industrial development of different economies.

8. Clarity & Language

• Overall, the language is clear, but some sections are overly technical. Simplifying jargon-heavy sentences, especially in the methodology and results sections, would help make the paper accessible to a broader audience.

• A few minor grammatical errors were noted, particularly with article usage and sentence length. Proofreading would help with flow.

Accept. Check the accuracy of words and sentences

9. Figures and Tables

• The figures (like the PMC surface maps) are useful but could be better labeled to guide interpretation. Adding more explanatory captions would help.

• The inclusion of high-frequency words in Table 1 is insightful, though this could be integrated into a more visual format (e.g., word clouds) to make it more engaging.

Accept. Add explanations before the PMC surface maps and add word clouds as figure 5.

Final Thoughts:

This is a strong paper with a solid methodology, but it would benefit from streamlining certain sections and emphasizing the real-world implications of the findings. The application of intelligent analysis and the PMC index is innovative, but the value it brings to policy-making could be more directly emphasized throughout the paper.

6. PLOS authors have the option to publish the peer review history of their article (what does this mean?). If published, this will include your full peer review and any attached files.

Do you want your identity to be public for this peer review? For information about this choice, including consent withdrawal, please see our Privacy Policy.

Reviewer #1: No

Reviewer #2: No

---

## [Decision Letter · Decision Letter 1]

9 Dec 2024

Research on the Evolution and Evaluation of Chinese Sports Industry Policies Based on Intelligent Analysis

PONE-D-24-43160R1

Dear Dr. Ma,

We’re pleased to inform you that your manuscript has been judged scientifically suitable for publication and will be formally accepted for publication once it meets all outstanding technical requirements.

Kind regards,

Tinggui Chen

Academic Editor

PLOS ONE

Additional Editor Comments (optional):

Reviewers' comments:

Reviewer's Responses to Questions

**Comments to the Author**

1. If the authors have adequately addressed your comments raised in a previous round of review and you feel that this manuscript is now acceptable for publication, you may indicate that here to bypass the “Comments to the Author” section, enter your conflict of interest statement in the “Confidential to Editor” section, and submit your "Accept" recommendation.

Reviewer #1: All comments have been addressed

Reviewer #2: All comments have been addressed

2. Is the manuscript technically sound, and do the data support the conclusions?

Reviewer #1: Yes

Reviewer #2: Yes

3. Has the statistical analysis been performed appropriately and rigorously? 

Reviewer #1: Yes

Reviewer #2: Yes

4. Have the authors made all data underlying the findings in their manuscript fully available?

Reviewer #1: Yes

Reviewer #2: Yes

5. Is the manuscript presented in an intelligible fashion and written in standard English?

Reviewer #1: Yes

Reviewer #2: Yes

6. Review Comments to the Author

Reviewer #1: All my concerns have been addressed. It can be accepted with minor grammatical and sentence structure correction.

Reviewer #2: I would like to say many thanks for addressing the comments that I recommended. No I recommend to publish

7. PLOS authors have the option to publish the peer review history of their article (what does this mean?). If published, this will include your full peer review and any attached files.

Reviewer #1: **Yes: **Asim Ahmad

Reviewer #2: No

---

## [Editor Report · Acceptance letter]

7 Jan 2025

PONE-D-24-43160R1 

PLOS ONE

Dear Dr. Ma, 

I'm pleased to inform you that your manuscript has been deemed suitable for publication in PLOS ONE. Congratulations! Your manuscript is now being handed over to our production team.

Kind regards, 

on behalf of

Dr. Tinggui Chen 

Academic Editor

PLOS ONE